# The Hepatitis B Virus Genotypes E to J: The Overlooked Genotypes

**DOI:** 10.3390/microorganisms11081908

**Published:** 2023-07-27

**Authors:** Rayana Maryse Toyé, Carmen Luisa Loureiro, Rossana Celeste Jaspe, Fabien Zoulim, Flor Helene Pujol, Isabelle Chemin

**Affiliations:** 1Institut National de la Santé et de la Recherche Médicale (Inserm) U1052, Centre de Recherche en Cancérologie de Lyon (CRCL), 151 Cours Albert Thomas, 69003 Lyon, France; rayanatoye@gmail.com (R.M.T.); fabien.zoulim@inserm.fr (F.Z.); 2Laboratorio de Virología Molecular, Centro de Microbiología y Biología Celular (CMBC), Instituto Venezolano de Investigaciones Científicas (IVIC), Caracas 1020A, Venezuela; cloureir@gmail.com (C.L.L.); rossanajaspe@gmail.com (R.C.J.); 3Collégium de Lyon, Institut d’Etudes Avancées, Université Lyon 2, 69007 Lyon, France

**Keywords:** hepatitis B virus, genotype E, genotype F, genotype G, genotype H, genotype I, genotype J, Africa, Latin America, Asia

## Abstract

Hepatitis B virus (HBV) genotypes E to J are understudied genotypes. Genotype E is found almost exclusively in West Africa. Genotypes F and H are found in America and are rare in other parts of the world. The distribution of genotype G is not completely known. Genotypes I and J are found in Asia and probably result from recombination events with other genotypes. The number of reported sequences for HBV genotypes E to J is small compared to other genotypes, which could impact phylogenetic and pairwise distance analyses. Genotype F is the most divergent of the HBV genotypes and is subdivided into six subgenotypes F1 to F6. Genotype E may be a recent genotype circulating almost exclusively in sub-Saharan Africa. Genotype J is a putative genotype originating from a single Japanese patient. The paucity of data from sub-Saharan Africa and Latin America is due to the under-representation of these regions in clinical and research cohorts. The purpose of this review is to highlight the need for further research on HBV genotypes E to J, which appear to be overlooked genotypes.

## 1. Introduction

Hepatitis B virus (HBV) is the prototype virus of the family *Hepadnaviridae*, a peculiar family of partially double-stranded deoxyribonucleic acid (DNA) viruses approximately 3200 nucleotides in length (3182–3248 nucleotides) encoding four genes: the viral polymerase (with reverse transcriptase activity), the envelope gene (which codes for three surface antigen proteins, depending on the start codon used), the pre-core/core gene, and the X gene, in overlapping reading frames [1,2].

HBV is classified into nine genotypes (A–I) and one putative genotype (J) with a genome-wide intergenomic sequence divergence of at least 7.5% [3,4]. Except for genotypes E, G, and putative J, all HBV genotypes are further subdivided into subgenotypes [5,6]. At least 30 subgenotypes (displaying a divergence of 4–7.5%) have been described, with distinct ethnic-geographic distribution and clinical outcomes [1,7]. The geographic distribution of HBV genotypes is highly associated with their regional host population, and with socio-demographic, ethnic, or migratory factors, and endemicity. Genotype A is found in North America, Europe, Southeast Africa, and India; genotypes B and C are found in Asia and Oceania; genotype D is the most widespread and is found in North America, North Africa, Europe, the Middle East, and Oceania; genotype E is found in West and Central Africa; genotype F is found in America, and genotypes G and H are found in Central and South America. Genotype I is found in Vietnam and Laos, and putative genotype J was reported in Japan [8,9,10]. The genomes of genotypes I and J are likely recombinants of genotype C, with unknown genotypes [7]. The phylogeny of genotype F is unusual compared to the other genotypes, with little intra-subgenotype diversity and yet long evolutionary distances between the six subgenotypes. The phylogenies of genotypes E, G, and H are relatively limited [1].

Humans have been infected with HBV for millennia. Analyses of HBV genomic data from Eurasians and Native Americans that lived between ~10,500 and ~400 years ago revealed that the most recent common ancestor of all HBV lineages existed between ~20,000 and 12,000 years ago [11]. Prehistoric lineages of HBV have undergone several replacements during human history. The only remnant of this prehistoric HBV diversity seems to be the rare genotype G, which is speculated to have re-emerged during the human immunodeficiency virus (HIV) pandemic or even before [11,12].

In a recent analysis including all extant HBV sequences, McNaughton and colleagues analyzed the 248 sequences they found available for genotypes E to I. As documented previously, there is considerable variation in the number of sequences available for each genotype, with genotypes E through I being relatively neglected. This variation may highlight the under-sampling of HBV sequences in several regions (particularly resource-limited regions) [1] (Figure 1).

Most of the information available on HBV virology and pathogenicity has been generated with the first discovered human HBV genotypes A–D, while the remaining genotypes have seldom been studied. In fact, while the HBV database (HBVdb) (accessed on 26 March 2023) contains almost 7500 whole-genome sequences of HBV genotypes A to D, less than 720 sequences (<10%) were available for HBV genotypes E to H, and no sequences were available for genotypes I and J.

The purpose of this review was to highlight the data regarding HBV genotypes E through J and the need for further research on these overlooked genotypes.

## 2. The African Genotype E

Discovered relatively recently and described in 1992, HBV genotype E is the predominant genotype in West Africa, the only major region in the world where HBV is hyperendemic [4,5,14,15,16,17,18,19]. As some authors have pointed out, genotype E has not been studied as thoroughly as the other genotypes, likely due to its predominantly African distribution [20]. This genotype is estimated to be involved in nearly 20% of chronic HBV cases worldwide, although this estimate may be biased because of the limited data in Africa. HBV genotype E is rarely found outside of Africa, except in people of African descent, and has very low nucleotide divergence (1.2–1.75%). These observations suggest a recent introduction of this genotype into the human population [5,21].

Genotype E is 3212 nucleotides in length and has a single three-nucleotide deletion in the preS domain. This genotype is also characterized by a unique serological subtype *ayw4* and a unique feature due to the introduction of another start codon Met^83^ in the preS1 region [22]. Only two lineages [20,23] and several regional clusters [21] have been reported for the dominant genotype in West Africa, genotype E. Indeed, in a recent phylogeography study of the complete HBV genotype E genomes, several West African countries were included in the analysis. However, some countries (Côte d’Ivoire, the Gambia, Guinea Bissau, Mali, and Mauritania) were not included due to the lack of published sequence data for these countries, which led to a huge gap in the current classification. In the present study, our HBV genotype E analysis included two sequences from Côte d’Ivoire, but no sequences from the other countries mentioned above or from Senegal. The phylogenetic analysis performed in our study highlighted the two lineages described above, although some sequences did not fall into these lineages (Figure 2).

The need for large-scale investigations of HBV genotype E, considering the neglected HBV genotypes, has recently been emphasized [5].

### 2.1. Origin of the African HBV Genotype E

The first description of genotype E in a Cameroonian blood donor occurred in 1992 [5,19]. To date, 97% of genotype E cases are in sub-Saharan Africa, and it is estimated that this genotype is responsible for around 17% of all HBV infections worldwide [5,26]. Only sporadic cases of genotype E have been described outside the African continent despite the migrations linked to the slave trade from West Africa to the New World. Hence, this genotype may not have circulated at that time and most likely appeared after the end of the slave trade [5].

The West African origin of genotype E has been suggested by Ingasia and collaborators in their recent phylogeography study, showing strain dispersals from there to the European region. Recent discoveries, including detection of this genotype in isolated African tribes (Pygmies and Bantus) and from skeletal remains from the Neolithic and Bronze era, support the theory that HBV genotype E existed before and was reintroduced recently [5,27,28,29,30]. Moreover, HBV genotype E isolates from individuals without any link to the African continent such as Colombians [13] and native Indians [31], and from individuals from the Khoi San, have been reported [21]. Genotype E strains were shown to circulate in an isolated Afro-Colombian community, with no recent contact with Africa, for the first time in 2010. The authors therefore hypothesized that the virus spread with slaves brought to Colombia and has been circulating in this population since, or that the virus was introduced more recently due to the contact of its inhabitants with Africans [13].

The genotype spread rapidly in African regions, probably in association with a change in the route of transmission. It has been suggested that an abrupt change in the transmission pathway including contaminated vaccines may have caused the outbreak [21]. As a matter of fact, mass vaccination programs that took place in Western Africa appear to be a probable cause for its reintroduction [5]. Alternative transmission modes such as socio-cultural practices (scarification, traditional shaving) have also been suggested. Furthermore, the explosive transmission and spread of this genotype in West Africa might be related to the onset of the HIV epidemic in the 1950s in this region with high rates of HBV and HIV co-infection [32]. However, it is still unclear why genotype E predominated over the historical African genotype Aa or A1, which originated from Africa and spread to the Americas through slavery [21]. There is no evidence indicating that immunity to genotype A (e.g., through vaccines) is less effective against genotype E as well [33].

HBV genotype E sequences were reported to be related to genotype D and chimpanzee strains. The authors suggested that this association might be explained by the probable descent from West Africa of the “group E” and chimpanzee strains [34]. HBV genotypes D and E are very close, and the relationship between the two genotypes has been questioned since the first phylogenetic studies on HBV. These questions have been clarified, and genotype E isolates were shown to cluster together and away from genotype D [35]. Genotype E, mostly exclusively found in Africans, was also isolated in chimpanzees. However, it remains uncertain whether it was transmitted by a human during captivity or before by another chimpanzee infected with a genotype E strain meaning that this strain might have been shared by humans and chimpanzees in Africa for an extended period. Moreover, the authors hypothesized that HBV genotype E might have once been a strain of the chimpanzee hepadnavirus and that its spread from chimpanzees to humans, or vice versa, might have occurred as a later event [36].

### 2.2. Pathogenicity of the Overlooked HBV Genotype E

There is a serious lack of data regarding the natural history of infection with genotype E, most of the evidence arising from empirical evidence. A limited number of studies have focused on the identification of the modes of transmission of this genotype [5]. Data on the clinical and virological characteristics, evolution of disease, and outcomes and treatment efficacy in patients infected with genotype E are scarce. The reported clinical characteristics of HBV genotype E are high viral load, high rates of HBeAg positivity, higher rates of chronicity compared to other genotypes [5], and perinatal transmission [22]. Several risk factors potentially influencing genotype E maternofetal transmission have been identified in pregnant women with chronic HBV infection, including viral load, HBV serological profile, duration of chronic infection, genotype, and mutations in the basal core promoter (BCP)/preCore (PC) or S regions (reviewed in [37]). A review published in 2014 reported that genotype E was the most difficult genotype to treat with a longer period required for treatment. Moreover, patients with genotype E displayed a lower mean decrease in hepatitis B surface antigen (HBsAg) levels at the end of treatment and a rebound of HBsAg during follow-up [38].

Although data are limited for this genotype, some variants and mutants have been reported. These may impact detection, vaccine response and disease severity. For example, the preS2 F22L mutation, described in Sudanese hepatocellular carcinoma (HCC) cases infected with genotype E, is associated with the development of HCC [5]. In addition, a study conducted in 2020 on samples from the Gambia showed the presence of new HBV variants characterized by a deletion in the preS2 region (preS2∆38–55). Those variants were likely associated with an increased risk of HCC in genotype-E-infected patients [39]. A study conducted in HIV and HBV co-infected Ghanaian patients showed a high proportion of drug resistance mutations in antiretroviral treatment (ART)-experienced patients, and the majority of those with resistance mutations harbored HBV genotype E [40].

A number of recombinant strains of genotype E with other HBV genotypes, mostly A and D co-circulating within Africa, have been reported [5]. However, the scarcity of A/E recombinants does not suggest a long-term co-circulation of both genotypes, nor their high prevalence. A probable scenario is that HBV was relatively rare in Africa until the massive spread of genotype E by a new route of transmission [33].

A carcinogenic potential of genotype E has been suggested in African regions where endemicity of this genotype is correlated with higher rates of HCC. Probable mechanisms underlying the carcinogen potential of HBV genotype E include dietary cofounders (aflatoxin), viral (HIV co-infection), or immune escape events [5], highlighting the need for further research on the subject.

## 3. The American Genotypes F and H

HBV genotypes F and H are the most divergent genotypes of all human HBV genotypes. Their complete genome exhibits around 15% nucleotide divergence from the sequences of the other human genotypes. In the study of McNaughton et al. [1], neither of these two genotypes exhibited deletions and insertions; thus, their genome is 3215 nucleotides in length [1]. Six subgenotypes have been described for HBV genotype F (F1–F6) [41]. In addition, different clades have been described for subgenotype F1 (F1a, b, c, and d) and F2 (F2a and b). Two clades have been proposed for HBV genotype H [42].

American HBV genotypes are found on all the continents, though their prevalence is very low outside America, and it is particularly high in indigenous populations of Latin America. The relative frequency of HBV genotypes F or H is closely correlated with the degree of admixture of the general population with Amerindians [43]. In countries with a lower frequency of Amerindian genes, but where the contribution of African genes is even lower, such as Argentina, Chile, Colombia, and Venezuela, HBV genotype F is predominant. In contrast, in Brazil, where the Amerindian contribution is lower than the African one, HBV genotype F is not predominant, even in indigenous populations. In the United States of America (USA) and Canada, HBV genotype F is frequently found among Alaska Natives [44].

The distribution of the American HBV genotypes and subgenotypes is also related to the early migrations of the first settlers of the Americas. HBV genotype F1 is found on the West coasts of the continent, paralleling the transatlantic route of the first American settlers [45]. Subgenotype F1b is found in Alaska but also in Peru and the South Cone (Chile, Argentina, and Uruguay). Subgenotypes F1a and F1c are mainly found in Central America. Subgenotype F3 has been described mostly in Colombia and Venezuela, subgenotype F2a in Venezuela and Brazil, and subgenotype F2b only in Venezuela and Martinique. Subgenotype F4 is found in Argentina and Bolivia but also in Brazil. Subgenotypes F5 and F6 are less frequently mentioned in the literature. Two complete genome sequences have been identified for HBV subgenotype F5, all from the western region of Panama [46], and five isolates for subgenotype F6, all but one from Argentina [45]. The six subgenotypes of HBV genotype F and the clades of these subgenotypes are shown in Figure 3.

Genotype H is mainly found in Mexico and Nicaragua, but has also been reported in the USA and Japan [42,43,47]. Two complete genotype H genome sequences from Nicaragua form a separate branch in the phylogenetic tree of HBV non-recombinant genotype H isolates, with a mean distance of almost 3% between the two clades, somehow lower than the one accepted for subgenotype assignment, suggesting an ongoing subgenotype diversification. Indeed, as previously mentioned, two clades have been proposed for this genotype: clade I may have spread in Mexico and Nicaragua around the 1960s–1970s, and clade II disseminated to other American and Asian countries around one decade later. However, analysis of a larger number of sequences shows that the two clades of HBV genotype H are not clearly visible (Figure 4). The phylogeographic analysis suggests a Mexican origin for this genotype [42].

### 3.1. Are the American Genotypes the Ancestral Lineage?

The American genotypes being the most divergent of human genotypes, they have been suggested to be at the origin of all human genotypes [48]. Phylogenetic analyses advocate for human HBV (genotype C specifically) originating in the Old World and its spreading following prehistoric human migrations [49]. Co-speciation of human HBV and their primate hosts, both in the Old and the New World, have also been proposed [50]. The identification of bat hepadnaviruses, in the Old and the New World, implies that these viruses might have played role in cross-species transmission. None of the proposed hypotheses alone adequately explain the current geographical distribution of human HBV genotypes [51]. The main difficulty in reconstructing the evolution of HBV is the lack of consensus concerning viral substitution rates, which seems to be due to the alternative use of different calibration approaches. The small size and compact nature of the HBV genome also impair such evolutionary studies [43].

A previous study suggested an accelerated substitution rate for the American genotypes F and H [52]. In contrast, another study found no such evidence: the American genotypes F and H most likely represent the sister lineages of all other human and ape HBV genotypes [53]. Another group reconstructed the evolution of HBV genotype F with the working hypothesis of a co-divergence with humans in Central and South America. Subgenotype F1 may have spread along the Pacific coast and evolved in association with Central American and Andean cultures from the west of the continent. Subgenotypes F2–F6 spread along the Atlantic coast and through inner pathways. The selection of differential biological characteristics in these two main groups might be related to their evolution in host populations with different genetic backgrounds and demographic conditions [45].

Although the American genotypes exhibit a long-term evolution within the American human settlers [45], Bayesian coalescent analysis estimated the time to the most recent common ancestor of the present HBV F subgenotype after the year 1000. The HBV F lineage was found to date back to the year 1040. The most ancient subgenotype might be F2 (time of the most recent common ancestor, tMRCA 1245), followed by F4 (tMRCA 1381), F1 (1650), and F3 (1730). The overall population infected with HBV genotype F increased in the 18th century and followed an initial expansion outward from Venezuela to other countries in Latin America. Thus, although HBV genotype F originated thousands of years ago, circulating strains of HBV genotype F seem to have spread in recent centuries, particularly in the 18th and 19th centuries [54].

HBV genotype H seems to have emerged more recently, with a tMRCA in 1933. Two clades have recently been proposed, the first composed of sequences from Central and North America and the other with sequences from North and South America, and Asia. However, as stated before, their divergence does not reach the 4% minimum to propose two subgenotypes [42].

### 3.2. Differential Pathogenicity of the American Genotypes

It is known that HBV genotypes differ in their pathogenic properties, including the risk of chronic infections and the ability to induce HCC [43]. As stated before, HBV genotype C is one of the genotypes most clearly associated with an increased risk of HCC development. However, HBV genotype F has also been associated with an increased risk of HCC generation. The most complete studies on the association of HBV genotype F and the early development of HCC come from Alaska, where this genotype is associated with a risk factor of HCC development even higher than genotype C, and at a younger age [55].

The same association between HBV genotype F and cancer at a young age has been described in Peru [56]. Other factors may predispose these American autochthonous populations to the development of HCC. A peculiar pattern of global DNA hypermethylation has been observed in Peruvian tumors, associated with one of the four Native American mitochondrial haplogroups A–D, haplogroup B, in sharp contrast with the usual pattern of hypomethylation observed in HCC [57]. In addition, it is known that mycotoxin is an important cofactor for the development of HCC. Although the consumption of aflatoxin does not seem to be as frequent as in Africa, the Peruvian population is exposed to significant concentrations of other mycotoxins, such as fumonisins [58]. Independently of other cofactors, HBV genotype F seems to be associated in some conditions with a rapid and frequent development of HCC. The subgenotype of HBV associated with HCC in both settings (Alaska and Peru) is F1b [43]. There was no evidence of co-infection with hepatitis delta in either setting.

This subgenotype, and probably F2, was shown to be associated with a higher frequency of HCC development. They have also been associated with a higher frequency of mutations in their genomes, known to display a correlation with HCC presentation, such as a mutation in the BCP and preS deletions [43]. However, not all American genotypes are prone to developing HCC. HBV subgenotypes F3 and F4 exhibit a lower frequency of BCP mutations and preS deletions [43]. Nevertheless, very few data are available concerning HCC in Amerindians, and liver cancer may be underdiagnosed in many of those populations, where the American genotypes are frequent and highly endemic [43].

HBV genotype H has rarely been associated with HCC [43]. Patients infected with this genotype usually are asymptomatic without displaying liver disease, often with acute liver damage modest or undetectable, and common occult B infection (OBI). In chronic patients, HBV genotype H is usually associated with the presence of other co-morbidities, such as alcoholism, obesity, and co-infection with hepatitis C virus (HCV) or HIV, and low prevalence of HCC [59].

## 4. The Enigmatic Genotype G: Also an American Genotype?

HBV genotype G has been called an enigmatic genotype, because it is a rare type, for which the biology is poorly understood [12,60,61]. After its discovery in France and the USA, genotype G was later found in other countries of the world, particularly in all the Americas (Figure 1).

Many reported genotype G isolates have numerous mutations in the BCP and preC region, and a unique 36-basepair insertion downstream of the core start codon. HBV genotype G possesses the longest genome of the human HBV genotype, with 3248 nucleotides [1]. The pre-core mutations preclude the expression of HBeAg, while the BCP mutations may enhance replication. This insertion may increase the expression of the core [62]. HBV genotype G might not be prone to persist as a chronic infection and has frequently been reported in combination with other HBV genotypes [60] and alone [59,63,64].

Despite its wide distribution, genotype G exhibits low genetic diversity [12], as shown in Figure 5. This low genetic diversity might suggest a recent re-emergence after a long period at a low level of persistence [11,60]. However, more recent phylodynamic studies suggest a tMRCA in 1855 probably in the USA, and its dissemination to other continents (South and Central America, Europe, Asia, and Africa) more than one century later (around the 1970s) [12]. Other studies suggest an association with HBV genotype E and a re-emergence of this genotype in Central Africa at the beginning of the past century [65].

This genotype has frequently been detected in HIV-positive patients, and phylodynamic patterns have highlighted a sharp increase in its dissemination co-occurring with the HIV pandemic, possibly associated with highly sexually active groups and users of injectable drugs [66]. Also, parenteral transmission has been reported in blood donors and hemodialysis patients. Phylogenetic analyses suggest that HBV genotype G has not co-evolved with any of the other genotypes. However, there may be an evolutionary association with HBV genotype H, because of their similar routes of transmission, especially among high-risk groups, like men who have sex with men (MSM) and illicit drug users [12].

With respect to pathogenicity, in HIV-HBV co-infected patients, HBV genotype G was shown to be an independent risk factor for liver fibrosis progression [67]. However, a previous study failed to identify an association between HBV genotype G infection in HIV co-infected patients and progression to liver fibrosis [68].

## 5. The Asian Genotypes I and J

Genotypes I and J are the most recently described HBV genotypes and originate predominantly in Asia. Genotype I was first isolated in 2008 from a Vietnamese patient [8] and has since been described in other countries including Laos, India, and China [9,69,70]. This genotype is 3215 nucleotides in length and has been subdivided into two subgenotypes, I1 and I2, with distinct serological subtypes (*adw2* and *ayw2*) but with an intergroup distance below the 4% threshold for subgenotype classification. HBV genotype I is the result of recombination events between genotypes A, C, and G and has been considered an indeterminate genotype clustering close to genotype C in whole-genome analysis and with genotype A in polymerase (reviewed in [4,26,71]). In a scoping review conducted in 2018, the authors pointed out that, although the data confirm that genotype I was rare and present only in Southeast Asia, its frequency may have been underestimated due to missed genotype I infections in previous studies [26]. A recent systematic review of the prevalence of HBV genotypes and subtypes in Asia showed that HBV genotype I was present in all Asian countries, albeit at a low frequency (less than 5%) in each [72]. In addition, it has been reported that perinatal transmission is frequent in regions where HBV genotype I prevails [4].

Genotype J was isolated in 2009 from a Japanese case of HCC who resided in Borneo [4,10,73]. HBV genotype J is the shortest of the HBV genotypes along with genotype D, with 3182 nucleotides in length and a 33-nucleotide deletion in the preS1 region. This genotype clusters with gibbon and orangutan genotypes but is closely related to an isolate of HBV genotype C. The authors hypothesized that this patient’s route of infection was probably zoonotic or originated from human inhabitants of the island of Borneo [10]. In an analysis conducted by another group later, using more gibbon and orangutan HBV sequences, it was shown that the so-called genotype J was in fact a recombinant of genotype C and gibbon in the S region and may represent interspecies transmission. The authors suggested that before confirming the existence of this 10th genotype, additional sequences should be identified (reviewed in [4]).

These genotypes are still the subject of debate, as in 2016 they had not yet been ratified by the International Committee on Taxonomy of Viruses (ICTV) [73]. This is still the case for the putative genotype J [74], for which a single isolate has been identified.

## 6. Recombination in the Overlooked Genotypes

The characteristics (genome length, geographic distribution, frequency of recombination, and HCC propensity) of the genotypes described in this paper are summarized in Table 1. Recombination occurs relatively frequently in the HBV virus [6,75]. The neglected HBV genotypes do not escape this rule. The analysis of complete virus genomes available in the GenBank database in May 2022 showed the presence of recombinant isolates for genotypes E to H (Table 1). Interestingly, the frequency of recombination for HBV genotype G was significantly higher compared to the other genotypes (*p* < 0.001, Chi square test). This relatively high rate of recombination for HBV genotype G could be due to its frequent co-infection with other HBV genotypes, as stated above. Furthermore, regarding genotype I, it has been shown to be composed entirely of recombinant fragments [4]. In our analysis, when we searched GenBank for HBV genotype I sequences, we found only 16 records (6 I1 sequences and 10 I2 sequences). However, when we performed a BLAST analysis of the I sequence, we found over 100 similar sequences, indicating that these sequences may not be correctly identified in GenBank. As for the only existing sequence of HBV genotype J, it shows no recombination with human HBV sequences but is likely a human–primate HBV recombinant.

## 7. Is There a Need for a Vaccine Specific for the Overlooked Genotypes?

All HBV isolates share the “a” determinant, the main protein region of the HBsAg, against which neutralizing antibodies are mainly produced. However, since HBV genotypes F and H are the most divergent of the human genotypes, the concern has been raised several times of a likely reduced effectiveness for these genotypes, because of the use of a vaccine based on an HBV genotype A2 genome [76].

In 2011, a study described the detection by nucleic acid testing of six blood donors positive for HBV DNA, despite being vaccinated against HBV: five of them were infected with non-A2 genotypes, including one with an HBV genotype F1, rather uncommon in the USA [77]. Two other studies reported HBV infection with genotype F isolates, despite vaccination [78,79].

These reports suggested the need for evaluating the effectiveness of the HBV vaccine against other viral genotypes. An excellent review presented the effectiveness of vaccine worldwide, irrespective of the prevailing HBV genotype [76]. Of particular interest are the studies in indigenous South American populations, where the American genotypes prevail, and the vaccine has also proved to be effective [76,80,81,82,83].

In the case of HBV genotypes E and G, these genotypes are less divergent from the A2 genotype, so no concern has been raised on the effectiveness of the currently used HBV vaccines.

## 8. Conclusions

Genotypes and subgenotypes can influence the natural history of infection. Comparing different (sub)genotypes is often difficult as they do not circulate in the same populations. Regions in which genotype E is endemic are characterized by a higher incidence of HCC, and this has been confirmed in epidemiological studies on the carcinogenic potential of genotypes E and F in Alaska. Although the mechanisms underlying this oncogenic potential have not been deciphered, they could be related to the host immune response, and to other cofounders such as HIV co-infection, dietary iron overload, or mycotoxin consumption.

Horizontal transmission can occur mainly through HBeAg-positive family members or household contacts, playmates, or unsafe medical interventions. Very few studies have been carried out to identify routes of transmission of genotypes E and F. In the Gambia, mother-to-child transmission (MTCT) is responsible for 16% of chronic infections and increases the risk of persistent viral replication and severe liver disease [84].

As curative therapies are being developed, it will be important to monitor patients for progression to liver cancer, even if they have been cured of chronic hepatitis B (CHB) infection. The HBV genotype may influence the efficacy of antiviral therapies, although most studies that analyzed the role of HBV genotype in the treatment with nucleotide analogs (NA) mostly focused on genotypes A, B, C, and D. Lamivudine (LAM) is the NA which was used first in the world, and the association between HBV genotype and LAM has been demonstrated both in terms of response and the development of resistance mutations. Collectively, current international treatment guidelines do not consider patients infected with genotypes E to J. There is thus an urgent need for further large-scale investigations of overlooked HBV genotypes. In order to deliver proper medical care, improve knowledge on the response to treatment, and the development of resistance of relatively understudied genotypes like E and F, it is critical to adapt recommendations that could differ from those issued for other genotypes. To date, very few studies have focused on the response to vaccination of genotypes E, F, and G, although vaccination began over four decades ago.

In conclusion, genotypes E to J display unique origins and evolutions, as well as molecular and epidemiological characteristics. Their natural history has not been extensively studied, and information about the virological breakthrough as a result of vaccination is scarce. Consequently, it is important that CHB patients infected with genotypes E to J are included in clinical trials focusing on new antiviral therapies, biomarkers, and other possible preventive methods. Infections with genotypes E, F, and H occur in areas where HBV continues to be hyperendemic. In order to reach WHO targets for the worldwide elimination of viral hepatitis as a public health burden by 2030, there is an urgent need for more in-depth and large-scale investigations into genotypes E to J, which are under-represented in current studies.

## Figures and Tables

**Figure 1 microorganisms-11-01908-f001:**
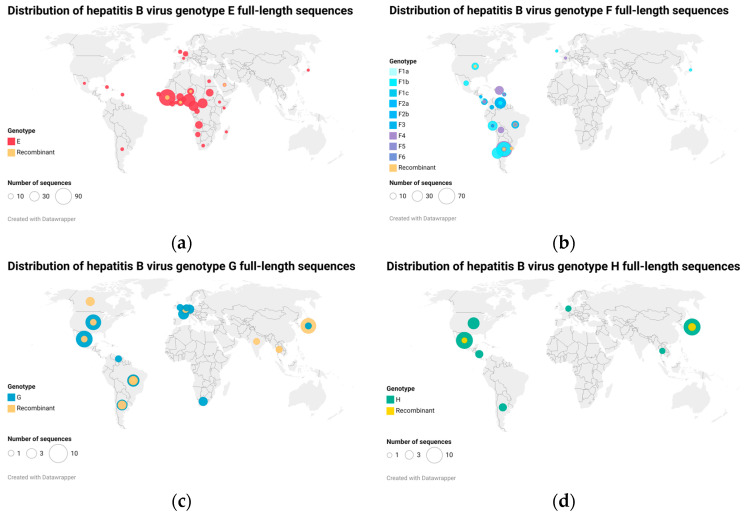
Maps of hepatitis B virus (HBV) whole-genome sequence distribution worldwide; (**a**) genotype E sequences, (**b**) genotype F sequences, (**c**) genotype G sequences, and (**d**) genotype H sequences. All available full-length sequences for HBV genotypes E to H were downloaded from GenBank and HBV database (HBVdb). The number of complete genome sequences of genotypes E, F, G, and H included in the map analysis (including recombinants) after removal of redundant sequences was 332, 339, 59, and 36, respectively. HBV genotype E has also been found in Afro-descendant populations in Colombia [13], but no complete genome is available.

**Figure 2 microorganisms-11-01908-f002:**
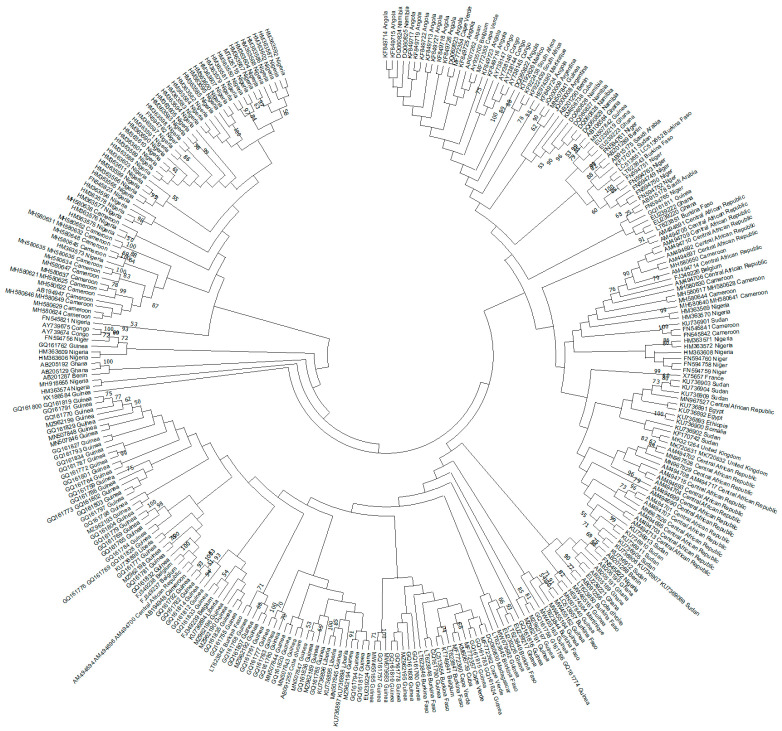
Phylogenetic analysis of HBV genotype E sequences. The evolutionary history was inferred by using the Maximum Likelihood (ML) method and General Time Reversible (GTR) model. The tree with the highest log likelihood is shown. The percentage of trees in which the associated taxa clustered together is shown next to the branches. Initial tree(s) for the heuristic search were obtained automatically by applying Neighbor-Join and BioNJ algorithms to a matrix of pairwise distances estimated using the Maximum Composite Likelihood (MCL) approach and then selecting the topology with the superior log likelihood value. A discrete Gamma distribution was used to model evolutionary rate differences among sites (3 categories). The rate variation model allowed for some sites to be evolutionary. The tree is drawn to scale, with branch lengths measured in the number of substitutions per site. This analysis involved 307 non-redundant nucleotide sequences out of the 326 sequences initially included (19 identical sequences had to be omitted). Codon positions included were 1st + 2nd + 3rd + Noncoding. Evolutionary analyses were conducted in Molecular Evolutionary Genetics Analysis (MEGA) version 11 [24]. Recombinant sequences were identified using the jumping profile Hidden Markov Model (jpHMM) tool [25] and were not included in the phylogenetic analysis, as for identical sequences. All the sequences are included in Appendix A.

**Figure 3 microorganisms-11-01908-f003:**
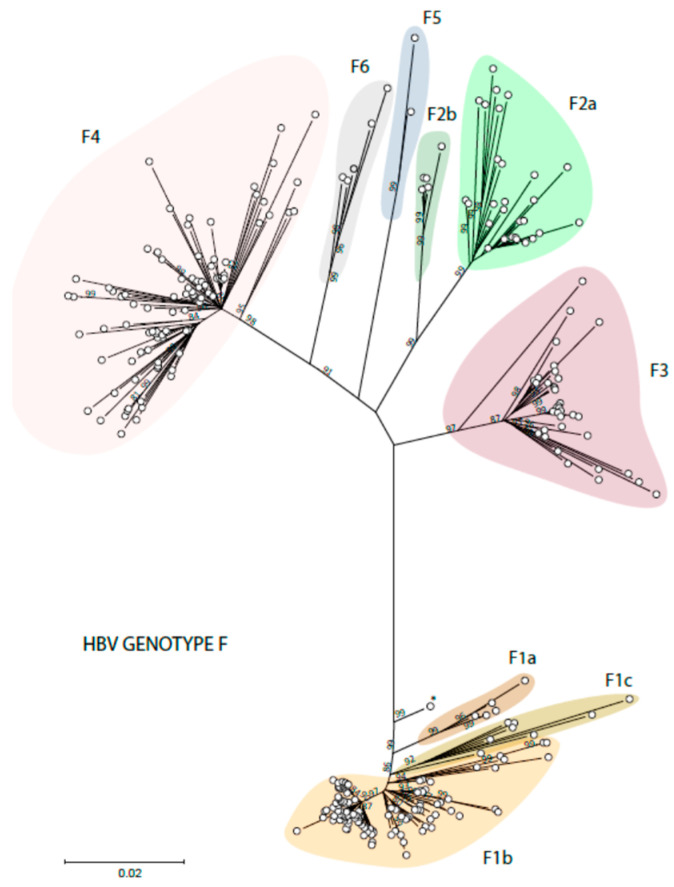
Phylogenetic analysis of HBV genotype F. The evolutionary history was inferred by using the ML method and GTR model. The tree with the highest log likelihood is shown. The percentage of trees in which the associated taxa clustered together is shown next to the branches. Initial tree(s) for the heuristic search were obtained automatically by applying Neighbor-Join and BioNJ algorithms to a matrix of pairwise distances estimated using the MCL approach and then selecting the topology with the superior log likelihood value. A discrete Gamma distribution was used to model evolutionary rate differences among sites (3 categories). The rate variation model allowed for some sites to be evolutionary. The tree is drawn to scale, with branch lengths measured in the number of substitutions per site. This analysis involved 268 non-redundant nucleotide sequences out of the 354 initially included (identical sequences had to be omitted). Codon positions included were 1st + 2nd + 3rd + Noncoding. Evolutionary analyses were conducted in MEGA 11 [24]. Recombinant sequences were identified using the jpHMM tool [25] and were not included in the phylogenetic analysis, as for identical sequences. All the sequences are included in Appendix A.

**Figure 4 microorganisms-11-01908-f004:**
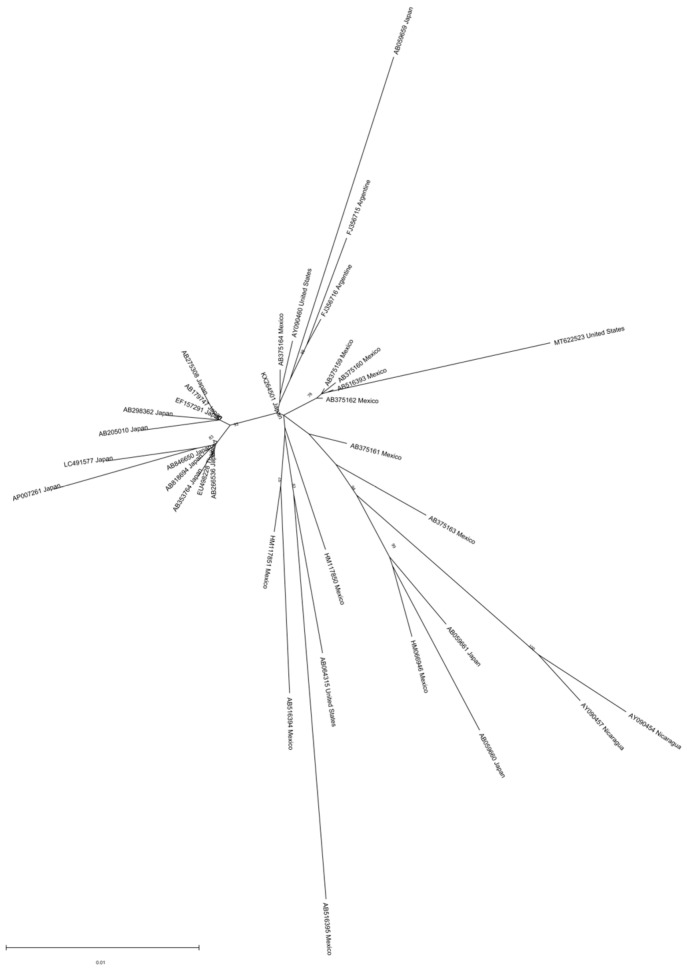
Phylogenetic analysis of HBV genotype H. The evolutionary history was inferred by using the ML method and GTR model. The tree with the highest log likelihood is shown. The percentage of trees in which the associated taxa clustered together is shown next to the branches. Initial tree(s) for the heuristic search were obtained automatically by applying Neighbor-Join and BioNJ algorithms to a matrix of pairwise distances estimated using the MCL approach and then selecting the topology with the superior log likelihood value. A discrete Gamma distribution was used to model evolutionary rate differences among sites (3 categories). The rate variation model allowed for some sites to be evolutionary. The tree is drawn to scale, with branch lengths measured in the number of substitutions per site. This analysis involved 39 nucleotide sequences. Codon positions included were 1st + 2nd + 3rd + Noncoding. Evolutionary analyses were conducted in MEGA 11 [24]. Recombinant sequences were identified using the jpHMM tool [25] and were not included in the phylogenetic analysis, as for identical sequences. All the sequences are included in Appendix A.

**Figure 5 microorganisms-11-01908-f005:**
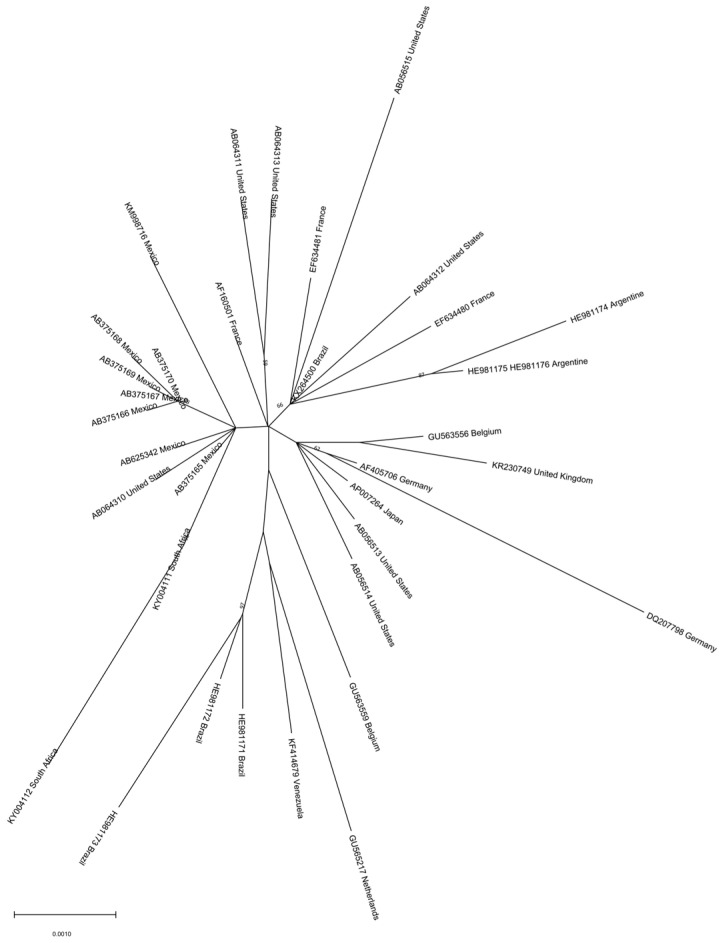
Phylogenetic analysis of HBV genotype G. The evolutionary history was inferred by using the ML method and GTR model. The tree with the highest log likelihood is shown. The percentage of trees in which the associated taxa clustered together is shown next to the branches. Initial tree(s) for the heuristic search were obtained automatically by applying Neighbor-Join and BioNJ algorithms to a matrix of pairwise distances estimated using the MCL approach and then selecting the topology with the superior log likelihood value. A discrete Gamma distribution was used to model evolutionary rate differences among sites (3 categories). The rate variation model allowed for some sites to be evolutionary. The tree is drawn to scale, with branch lengths measured in the number of substitutions per site. This analysis involved 35 nucleotide sequences. Codon positions included were 1st + 2nd + 3rd + Noncoding. Evolutionary analyses were conducted in MEGA 11 [24]. Recombinant sequences were identified using the jpHMM tool [25] and were not included in the phylogenetic analysis, as for identical sequences. All the sequences are included in Appendix A.

**Table 1 microorganisms-11-01908-t001:** Characteristics of the overlooked genotypes E to J of hepatitis B virus (HBV).

Genotype	Genome Size * (Nucleotides)	Subgenotypes	Geographic Distribution	Recombination **	Hepatocellular Carcinoma (HCC) Propensity
E	3212	Two lineages described	Africa	8/405 (2%)	Variants with preS2 deletions prone to HCC
F	3215	F1ad-F2ab-F3-F4-F5-F6	Central and South America, Alaska	13/354 (3.6%)	F1 and probably F2 more prone to HCC than F3 and F4
G	3248	No	America, Europe, Japan	24/59 (41%) ***	Unknown
H	3215	Two lineages described	Central America	3/39(8%)	Low
I	3215	I1-I2	Laos, Vietnam	Recombinant	Unknown
J	3182	Only one isolate	Japan (origin Borneo?)	Recombinant	Unknown

* The genome size of the prototype genome is shown. Different sizes found in other isolates are shown in Appendix A. ** The presence of recombinant isolates in full-length sequences was assessed using the jpHMM tool. *** A significantly higher frequency of recombination was found: *p* < 0.001.

## Data Availability

The data analyzed in this study are publicly available in GenBank and HBVdb databases using the accession numbers provided in Appendix A.

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
