# Peer review of "The Hepatitis B Virus Genotypes E to J: The Overlooked Genotypes"

_microorganisms, 2023, doi:10.3390/microorganisms11081908_

Round 1
Reviewer 1 Report
The paper is interesting and well written. The presented review article provides a comprehensive overview of the issue, is supported by a number of relevant citations, and draws adequate conclusions based on the sources. Overall, I find the paper very useful and recommend its publication.
The manuscript comprehensively addresses the issue of hepatitis B virus E-J genotypes. Most studies have focused on genotypes A and D (dominant in Europe and North America) and B and C (dominant in Asia). E-J genotypes are more common in countries with limited research resources and there are relatively few clinical studies focusing on these genotypes. Even relevant international guidelines and recommendations often pay insufficient attention to them.
The present manuscript (review) summarizes the current knowledge on the E-J genotypes of HBV and thus fills a certain gap in the literature. It analyses both epidemiological and clinical knowledge on these genotypes. It discusses their origin, distribution, diversity, mode of transmission, development of chronic disease, incidence of HCC or effectiveness of vaccination.
The paper is interesting and well written. The presented review article provides a comprehensive overview of the issue, is supported by a number of relevant citations, and draws adequate conclusions based on the sources.
I have only one suggestion for possible improvement. For a better quick overview the authors might consider adding a table that summarizes basic information available on the individual genotypes (E-J) addressed in the paper (e.g., geographic prevalence, genome size, prevalence of subtypes, degree of risk of progression to cirrhosis, development of HCC, or escape from vaccine protection).
Overall, I find the paper very useful and recommend its publication.
Author Response
Dear Editors,
We would like to thank the reviewers for their carefully and constructive comments, these undoubtedly improved our manuscript. We have addressed each comment below.
We also wish to thank the editor for reviewing the revised manuscript.
With best wishes,
The authors
Manuscript ID: microorganisms-2480284
Title: The hepatitis B virus genotypes E to J: the overlooked genotypes.
Authors: Rayana Maryse Toyé et al.
- Response to reviewer #1:
- General comments
The paper is interesting and well written. The presented review article provides a comprehensive overview of the issue, is supported by a number of relevant citations, and draws adequate conclusions based on the sources. Overall, I find the paper very useful and recommend its publication.
The manuscript comprehensively addresses the issue of hepatitis B virus E-J genotypes. Most studies have focused on genotypes A and D (dominant in Europe and North America) and B and C (dominant in Asia). E-J genotypes are more common in countries with limited research resources and there are relatively few clinical studies focusing on these genotypes. Even relevant international guidelines and recommendations often pay insufficient attention to them.
The present manuscript (review) summarizes the current knowledge on the E-J genotypes of HBV and thus fills a certain gap in the literature. It analyses both epidemiological and clinical knowledge on these genotypes. It discusses their origin, distribution, diversity, mode of transmission, development of chronic disease, incidence of HCC or effectiveness of vaccination.
The paper is interesting and well written. The presented review article provides a comprehensive overview of the issue, is supported by a number of relevant citations, and draws adequate conclusions based on the sources.
Authors’ response
The reviewer summarized our work, highlighting its relevance and recommending its publication.
- Major comment
I have only one suggestion for possible improvement. For a better quick overview the authors might consider adding a table that summarizes basic information available on the individual genotypes (E-J) addressed in the paper (e.g., geographic prevalence, genome size, prevalence of subtypes, degree of risk of progression to cirrhosis, development of HCC, or escape from vaccine protection).
Authors’ response
We found the reviewer’s suggestion on adding a table to summarize information on genotypes included in the manuscript highly relevant. The table has therefore been added (lines 453-457).
- Conclusions
Overall, I find the paper very useful and recommend its publication.
Authors’ response
We thank the reviewer for this comment.
Reviewer 2 Report
This review gives a brief overview of hepatitis B virus genotypes E to J, with the authors aiming to highlight the need for further research on these neglected genotypes.
The authors emphasize the lack of data for studies involving these genotypes. It would be interesting to make a more substantial comparison to the volume of studies described in the literature for the other genotypes A-D to possibly highlight this disparity. For example, comparing the number of sequences available in the public databases for each genotype.
The authors have included figures of phylogenetic trees of the different HBV genotypes but have not included any discussion or reference to these in the paper. The figures thus do not add anything to the manuscript in the current form.
This review lacks relevance as in many instances the authors reference other reviews, as opposed to the source data. There are also many instances where the authors have included statements taken from the references, with minor re-wording. For example, the sentence introducing genotype G (lines 336-337) is slightly modified from the reference Araujo et al. Front Microbiol 2022 (reference 45 in this manuscript).
In addition, there are a number of clarifications and issues, especially with references used, that need to be corrected as outlined in the specific comments to authors.
Specific comments:
Introduction:
Line 43. The authors describe genotype D as the “most common” genotype. This is in-accurate, and would be better described as “most widespread”. (Genotypes B and C are probably most common worldwide in terms of prevalence.
Line 47, The references given for the geographic locations for genotypes I and J are incorrect. Reference 5 refers to genotype E, and Reference 8 refers to genotypes found in Cuba.
Line 55. The statement that the “most recent common ancestor of all HBV lineages existed between ~20,000 and 12,000 years ago” needs a reference, probably reference 9.
Line 58. The two references used here are in the wrong order. Reference 10 should be used for the HIV pandemic statement, and reference 9 for the “or even before” statement.
Figure 1: This figure requires more information. Where were the sequences obtained from? What were the search parameters used to select the sequences included? How many sequences were included? The text mentions 248 sequences analyzed in reference 1, and the HBVdb has <720 sequences for genotypes E to H. Were either of these sets of sequence data used for this figure?
2. Genotype E:
Line 80. Reference 12 is incorrect here as this reference does not include any mention of genotype E.
Line 82. Could the authors please clarify if the statement “This genotype is estimated to be involved in nearly 20% of chronic HBV cases,” refers specifically to Africa, or is this worldwide?
Figure 2: The authors need to describe how the sequences used to create this figure were selected. The figure legend indicates there were 305 sequences involved, however the Supplemental Table 1 has 326 sequences listed.
Line 116. Could the authors please supply a reference for this statement?
Lines 117, 118. This statement is incorrect. The original source data (Velkov et al Genes, 2018. 9(10):495) states that 97% of genotype E is in sub-Saharan Africa, not that genotype E accounts for 97% of infections in sub-Saharan Africa. The authors need to include the original source, not a reference to a previous review.
Line 124. The first sentence in this paragraph requires a reference to the original source.
Lines 124-128. This sentence needs correction. Pygmies and Khoi San are African people, so have a connection to Africa. The reference included for this statement also is incorrectly worded.
Line 127. Please define “Indians”.
Lines 140-142. Please include the source reference for this statement (Jayaraman et al Transfusion, 2010).
Line 179. The authors have used the statement “in our team” in this sentence, however the reference given (Ref 24) does not include any of the same authors?
3. Genotypes F and H.
Lines 211-213. This statement should use the source reference (Livingston et al, Journal of Infectious Disease 2007) instead of referring to a previous review (reference 28).
Lines 221-222. This sentence “Subgenotypes F5 and F6 are frequently ignored in the literature.” Needs to be re-written. Perhaps “have not been described” or “are less frequently detected”.
Figure 3. Again, please describe how the sequences were selected for this analysis. Supplemental Table 2 contains 326 genotype F sequences, however the figure legend refers to the analysis involving 354 nucleotide sequences.
Line 240. The statement that two genotype H sequences from Nicaragua form a separate branch in the phylogenetic tree is not obvious from Figure 4. This statement is taken from reference 28 where the analysis makes this more apparent. This needs to be made clearer in the text.
Lines 265-268. Please include references for these statements. 1. “The American genotypes being the most divergent of human genotypes, they were suggested to be at the origin of all human genotypes.” 2. “Phylogenetic analyses advocates for human HBV…..following prehistoric human migrations.” And 3. “Co-speciation of human HBV…… have also been proposed.”
Line 273. Suggest the use of the word “alternative” instead of “improper”.
Lines 276-277. The study referred to hear (reference 33) found evidence there is accelerated substitution rate for HBV genotypes F and H. Please correct this sentence.
Line 300. The reference here should be reference 27, not 35.
Lines 314-320. One of the suggested reasons for the association with HCC at an early age can be co-infection with Hepatitis Delta virus in native South Americans. It would be helpful to include this co-factor in the discussion here.
4. Genotype G.
Lines 343-344. Please include a reference for this statement. “The function of the insertion is unknown……”
Figure 5. As for the other figures, please describe the selection of sequences used in this figure. There are 35 sequences in Supplemental Table 4, and there does not appear to be 59 sequences included in the figure, as stated in the figure legend.
5. Genotypes I and J.
Lines 384-385. Please include the original references describing the first genotype I isolations from these countries.
6. Recombination.
Lines 414-419. The data referring to the frequency of recombination across the genotypes would be better presented in a Table in the main text. The Supplemental tables 1-4 only contain lists of sequences from each genotype.
Line 427. The use of monkey” in “human-monkey HBV recombinant” is incorrect. HBV is known to infect different members of the great apes, not monkeys.
Lines 435-437. This sentence needs correction “five” should be “six” as the final sample described “genotype F1” is also a non-A2 genotype.
Lines 443-447. This statement needs to be re-worded. The study described is a clinical trial report showing safety of the vaccine based on 21 patients. This is not an example of the absence of requirements for new genotype-specific HBV vaccines.
Author Response
Dear Editors,
We would like to thank the reviewers for their carefully and constructive comments, these undoubtedly improved our manuscript. We have addressed each comment below.
We also wish to thank the editor for reviewing the revised manuscript.
With best wishes,
The authors
Manuscript ID: microorganisms-2480284
Title: The hepatitis B virus genotypes E to J: the overlooked genotypes.
Authors: Rayana Maryse Toyé et al.
- Response to reviewer #2:
- General comments
Comment 1
This review gives a brief overview of hepatitis B virus genotypes E to J, with the authors aiming to highlight the need for further research on these neglected genotypes.
Authors’ response
The reviewer summarized the objective of our work.
Comment 2
The authors emphasize the lack of data for studies involving these genotypes. It would be interesting to make a more substantial comparison to the volume of studies described in the literature for the other genotypes A-D to possibly highlight this disparity. For example, comparing the number of sequences available in the public databases for each genotype.
Authors’ response
We have taken note of the reviewer’s suggestion. Information regarding the number of sequences available for each genotype in the HBV database was already included in the manuscript (lines 74-77).
Comment 3
The authors have included figures of phylogenetic trees of the different HBV genotypes but have not included any discussion or reference to these in the paper. The figures thus do not add anything to the manuscript in the current form.
Authors’ response
We agree with the reviewer regarding the lack of description of the figures of phylogenetic trees. Consequently, they are now discussed in the manuscript as follows:
- Figure 2 (HBV genotype E): This sentence was included “The phylogenetic analysis performed in our study highlighted the two lineages described above, although some sequences did not fall into these lineages (Figure 2)” (lines 101-102).
- Figure 3 (HBV genotype F). This sentence was included “The six subgenotypes of HBV genotype F and the clades of these subgenotypes are shown in Figure 3” (lines 236-237).
- Figure 4 (HBV genotype H). This sentence was included “However, analysis of a larger number of sequences shows that the two clades of HBV genotype H are not clearly visible (Figure 4)” (lines 261-262).
- Figure 5 (HBV genotype G). The low genetic diversity has already been mentioned with reference to the figure (lines 367-368).
Comment 4
This review lacks relevance as in many instances the authors reference other reviews, as opposed to the source data. There are also many instances where the authors have included statements taken from the references, with minor re-wording. For example, the sentence introducing genotype G (lines 336-337) is slightly modified from the reference Araujo et al. Front Microbiol 2022 (reference 45 in this manuscript).
Authors’ response
As suggested by the reviewer, information concerning HBV genotype G has been edited.
“HBV genotype G has been called an enigmatic genotype, because it is a rare type, for which the biology is poorly understood [12,60,61]. After its discovery in France and the USA, genotype G was later found in other countries of the World, particularly in all the Americas (Figure 1).” (lines 355-358).
Comment 5
In addition, there are a number of clarifications and issues, especially with references used, that need to be corrected as outlined in the specific comments to authors.
Authors’ response
We thank the reviewer for her/his comments. We will try to address as best as possible the gaps and issues highlighted in the following answers.
- Specific comments:
Introduction:
Comment 6:
Line 43. The authors describe genotype D as the “most common” genotype. This is inaccurate and would be better described as “most widespread”. (Genotypes B and C are probably most common worldwide in terms of prevalence.
Authors’ response
As we agree with the reviewer, this information has been corrected in the manuscript (line 43).
Comment 7
Line 47, The references given for the geographic locations for genotypes I and J are incorrect. Reference 5 refers to genotype E, and Reference 8 refers to genotypes found in Cuba.
Authors’ response
We completely agree with the reviewer. The correct source references (8-10) for the geographic distribution of genotypes I and J have been included (line 47).
Comment 8
Line 55. The statement that the “most recent common ancestor of all HBV lineages existed between ~20,000 and 12,000 years ago” needs a reference, probably reference 9.
Authors’ response
As suggested by the reviewer, the reference has been included (line 55).
Comment 9
Line 58. The two references used here are in the wrong order. Reference 10 should be used for the HIV pandemic statement, and reference 9 for the “or even before” statement.
Authors’ response
The reviewer’s suggestion has been considered and the order of references corrected (line 58).
Comment 10
Figure 1: This figure requires more information. Where were the sequences obtained from? What were the search parameters used to select the sequences included? How many sequences were included? The text mentions 248 sequences analyzed in reference 1, and the HBVdb has <720 sequences for genotypes E to H. Were either of these sets of sequence data used for this figure?
Authors’ response
We agree with the reviewer on the need for more information on Figure 1. For clarification purpose, the sequences used in this study were retrieved from GenBank and HBV database. A total of 766 sequences (genotype E = 332, genotype F = 339, genotype G = 59 and genotype H = 36) were used for the maps in Figure 1, including sequences used in the phylogenetic analyses and recombinant sequences (excluded from phylogenetic analyses). This information has been added to the legend of Figure 1 (lines 66-70).
Regarding the 248 sequences from reference 1 mentioned in the manuscript, these were analyzed in a previous study by McNaughton and collaborators (genotype E = 145, genotype F = 80, genotype G = 3, genotype H = 11 and genotype I = 9). This information was mentioned for comparison purposes.
Regarding the number of ~720 sequences in the HBVdb, this was the number of sequences available in the database for genotypes E to H on 26 March 2023 (genotype E = 353, genotype F = 293, genotype G = 45 and genotype H = 27).
In our analyses, we used sequences available in HBVdb and GenBank, thus including the latter set of sequence data discarding duplicate sequences. Information on the source of sequences has been added to the legend of Figure 1 (lines 66-70).
- Genotype E:
Comment 11
Line 80. Reference 12 is incorrect here as this reference does not include any mention of genotype E.
Authors’ response
The reviewer’s comment has been considered. Several references have been included for genotype E origin since its first description to its West African predominance (line 83).
Comment 12
Line 82. Could the authors please clarify if the statement “This genotype is estimated to be involved in nearly 20% of chronic HBV cases,” refers specifically to Africa, or is this worldwide?
Authors’ response
As suggested by the reviewer, the statement has been clarified in the manuscript as follows: “This genotype is estimated to be involved in nearly 20% of chronic HBV cases worldwide” and a supplementary reference added (lines 86 and 89).
Comment 13
Figure 2: The authors need to describe how the sequences used to create this figure were selected. The figure legend indicates there were 305 sequences involved, however the Supplemental Table 1 has 326 sequences listed.
Authors’ response
We agree with the reviewer on the need for clarity regarding sequence selection. The 326 sequences represent the total number of sequences included in the analysis in first instance. However, 19 of these appeared to be identical, thus omitted in the phylogenetic tree which included 307 non-redundant sequences in the end. This information has been corrected in the legend of Figure 2 (line 113-114).
Comment 14
Line 116. Could the authors please supply a reference for this statement?
Authors’ response
As requested by the reviewer, references to the above-mentioned statement have been added to the manuscript (lines 123-124).
Comment 15
Lines 117, 118. This statement is incorrect. The original source data (Velkov et al Genes, 2018. 9(10):495) states that 97% of genotype E is in sub-Saharan Africa, not that genotype E accounts for 97% of infections in sub-Saharan Africa. The authors need to include the original source, not a reference to a previous review.
Authors’ response
The authors agree with the reviewer regarding this statement. This has been corrected in the manuscript and the original reference added (lines 124-126).
Comment 16
Line 124. The first sentence in this paragraph requires a reference to the original source.
Authors’ response
As requested by the reviewer, references have been included for the above-mentioned sentence (lines 133-136).
Comment 17
Lines 124-128. This sentence needs correction. Pygmies and Khoi San are African people, so have a connection to Africa. The reference included for this statement also is incorrectly worded.
Authors’ response
We agree with the reviewer regarding this sentence. Corrections have been made and original references added to the manuscript (lines 136-139).
Comment 18
Line 127. Please define “Indians”.
Authors’ response
As requested by the reviewer, the definition of Indians has been clarified in the manuscript (line 137). We meant a population originating from India (native Indians). More precisely, the source reference mentioned a population of North India (Haryana).
Comment 19
Lines 140-142. Please include the source reference for this statement (Jayaraman et al Transfusion, 2010).
Authors’ response
As requested by the reviewer, the source reference suggested for the above-mentioned statement has been included (lines 150-152).
Comment 20
Line 179. The authors have used the statement “in our team” in this sentence, however the reference given (Ref 24) does not include any of the same authors?
Authors’ response
Regarding this statement, the authors confirm that reference 24 (39 in the revised form) included two of the authors in the present review (F. Zoulim and I. Chemin): Cohen D, Ghosh S, Shimakawa Y, Ramou N, Garcia PS, Dubois A, Guillot C, Kakwata-Nkor Deluce N, Tilloy V, Durand G, Voegele C, Ndow G, Umberto d’Alessandro, Brochier-Armanet C, Alain S, Le Calvez-Kelm F, Hall J, Zoulim F, Mendy M, Thursz M, Lemoine M, Chemin I. HBV PreS2Δ38-55 variants: a newly identified risk factor for hepatocellular carcinoma. JHEP Reports 2020; 2:100144. However, since several teams are involved in this review, the statement has been removed from the manuscript (line 190).
- Genotypes F and H.
Comment 21
Lines 211-213. This statement should use the source reference (Livingston et al, Journal of Infectious Disease 2007) instead of referring to a previous review (reference 28).
Authors’ response
We agree with the reviewer for using the source reference for the above-mentioned statement. The reference has been included (line 224).
Comment 22
Lines 221-222. This sentence “Subgenotypes F5 and F6 are frequently ignored in the literature.” Needs to be re-written. Perhaps “have not been described” or “are less frequently detected”.
Authors’ response
As suggested by the reviewer, the sentence has been edited as follows: “Subgenotypes F5 and F6 are less frequently mentioned in the literature.” (lines 232-233).
Comment 23
Figure 3. Again, please describe how the sequences were selected for this analysis. Supplemental Table 2 contains 326 genotype F sequences, however the figure legend refers to the analysis involving 354 nucleotide sequences.
Authors’ response
We agree with the reviewer regarding inconsistency in the number of sequences. The information has been checked and after elimination of redundant sequences, the total number of sequences included in the phylogenetic tree was 268. This information has been corrected in the legend of Figure 3 (lines 247-248).
Comment 24
Line 240. The statement that two genotype H sequences from Nicaragua form a separate branch in the phylogenetic tree is not obvious from Figure 4. This statement is taken from reference 28 where the analysis makes this more apparent. This needs to be made clearer in the text.
Authors’ response
As recommended by the reviewer, this statement has been further discussed with the analysis of Figure 4, as indicated above (lines 261-262).
Comment 25
Lines 265-268. Please include references for these statements. 1. “The American genotypes being the most divergent of human genotypes, they were suggested to be at the origin of all human genotypes.” 2. “Phylogenetic analyses advocates for human HBV…..following prehistoric human migrations.” And 3. “Co-speciation of human HBV…… have also been proposed.”
Authors’ response
As requested by the reviewer, the references for the above-mentioned statements have been added (lines 279-284).
Comment 26
Line 273. Suggest the use of the word “alternative” instead of “improper”.
Authors’ response
As suggested by the reviewer, the word “improper” has been replaced by “alternative” (line 289).
Comment 27
Lines 276-277. The study referred to hear (reference 33) found evidence there is accelerated substitution rate for HBV genotypes F and H. Please correct this sentence.
Authors’ response
We agree with the reviewer. The sentence has been reedited as follows: “A previous study suggested an accelerated substitution rate for the American genotypes F and H [33]. In contrast, another study found no such evidence: the American genotypes F and H most likely represent the sister lineages of all other human and ape HBV genotypes [34].” (lines 291-294).
Comment 28
Line 300. The reference here should be reference 27, not 35.
Authors’ response
As we agree with the reviewer, the reference has been corrected (line 317).
Comment 29
Lines 314-320. One of the suggested reasons for the association with HCC at an early age can be co-infection with Hepatitis Delta virus in native South Americans. It would be helpful to include this co-factor in the discussion here.
Authors’ response
The reviewer’s comment has been considered. As we have not found information on this subject, a comment has been included (lines 338-339).
- Genotype G.
Comment 30
Lines 343-344. Please include a reference for this statement. “The function of the insertion is unknown……”
Authors’ response
The statement has been reedited as follows: “This insertion may increase the expression of the core.” and a reference has been added (lines 363-364).
Comment 31
Figure 5. As for the other figures, please describe the selection of sequences used in this figure. There are 35 sequences in Supplemental Table 4, and there does not appear to be 59 sequences included in the figure, as stated in the figure legend.
Authors’ response
We agree with the reviewer regarding inconsistency in the number of sequences. The number (35) has been corrected in the legend of Figure 5 (line 384).
- Genotypes I and J.
Comment 32
Lines 384-385. Please include the original references describing the first genotype I isolations from these countries.
Authors’ response
As requested by the reviewer, the original sources describing the first isolations of genotype I have been included in the manuscript (lines 404-406).
- Recombination.
Comment 33
Lines 414-419. The data referring to the frequency of recombination across the genotypes would be better presented in a Table in the main text. The Supplemental tables 1-4 only contain lists of sequences from each genotype.
Authors’ response
As the authors found the reviewer’s suggestion highly relevant, a table (Table 1) has been included in the manuscript with recombination data and other characteristics of the genotypes described in this paper (lines 453-457).
Comment 34
Line 427. The use of monkey” in “human-monkey HBV recombinant” is incorrect. HBV is known to infect different members of the great apes, not monkeys.
Authors’ response
We agree with the reviewer regarding the incorrect use of the word “monkey”. The term “monkey” has been substituted by “primate” (line 451).
Comment 35
Lines 435-437. This sentence needs correction “five” should be “six” as the final sample described “genotype F1” is also a non-A2 genotype.
Authors’ response
The reviewer’s suggestion has been considered. The sentence has been reworded to emphasize that the non-A2 sequences included a genotype F1 sequence which is rare in this area (lines 465-467). In fact, the five non-A2 genotypes were B2, C2, F1 (only one) and mixed/recombinant.
Comment 36
Lines 443-447. This statement needs to be re-worded. The study described is a clinical trial report showing safety of the vaccine based on 21 patients. This is not an example of the absence of requirements for new genotype-specific HBV vaccines.
Authors’ response
After considering the reviewer’s comment regarding the above-mentioned statement, the authors have decided to remove this example from the manuscript, along with the associated reference (lines 473-477).
Round 2
Reviewer 2 Report
This review gives a brief overview of hepatitis B virus genotypes E to J, with the authors aiming to highlight the need for further research on these neglected genotypes.
Thank you for satisfactorily addressing the comments/corrections outlined in the initial review.